# Engineering Commercial TiO_2_ Powder into Tailored Beads for Efficient Water Purification

**DOI:** 10.3390/ma15010326

**Published:** 2022-01-03

**Authors:** George V. Theodorakopoulos, Fotios K. Katsaros, Sergios K. Papageorgiou, Margarita Beazi-Katsioti, George Em. Romanos

**Affiliations:** 1Institute of Nanoscience and Nanotechnology, N.C.S.R. “Demokritos”, Ag. Paraskevi, 15310 Athens, Greece; f.katsaros@inn.demokritos.gr (F.K.K.); s.papageorgiou@inn.demokritos.gr (S.K.P.); g.romanos@inn.demokritos.gr (G.E.R.); 2School of Chemical Engineering, National Technical University of Athens, 9 Iroon Polytechniou Street, Zografou, 15780 Athens, Greece; katsioti@central.ntua.gr

**Keywords:** alginate, adsorption, batch reactor, commercial TiO_2_, shaping of photocatalysts, photocatalytic beads, water purification

## Abstract

In this study, efficient commercial photocatalyst (Degussa P25) nanoparticles were effectively dispersed and stabilized in alginate, a metal binding biopolymer. Taking advantage of alginate’s superior metal chelating properties, copper nanoparticle-decorated photocatalysts were developed after a pyrolytic or calcination-sintering procedure, yielding ceramic beads with enhanced photocatalytic and mechanical properties, excellent resistance to attrition, and optimized handling compared to powdered photocatalysts. The morphological and structural characteristics were studied using LN_2_ porosimetry, SEM, and XRD. The abatement of an organic pollutant (Methyl Orange, MO) was explored in the dark and under UV irradiation via batch experiments. The final properties of the photocatalytic beads were defined by both the synthesis procedure and the heat treatment conditions, allowing for their further optimization. It was found that the pyrolytic carbon residuals enabled the adhesion of the TiO_2_ nanoparticles, acting as binder, and increased the MO adsorption capacity, leading to increased local concentration in the photocatalyst vicinity. Well dispersed Cu nanoparticles were also found to enhance photocatalytic activity. The prepared photocatalysts exhibited increased MO adsorption capacity (up to 3.0 mg/g) and also high photocatalytic efficiency of about 50% MO removal from water solutions, reaching an overall MO rejection of about 80%, at short contact times (3 h). Finally, the prepared photocatalysts kept their efficiency for at least four successive photocatalytic cycles.

## 1. Introduction

Advanced Oxidation Processes (AOPs) are considered state-of-the-art water treatment technologies [1,2]. The objective of AOPs is the in situ generation of highly reactive species such as the hydroxyl radical (OH^•^), the superoxide anion radical (O_2_^−•^), hydrogen peroxide (H_2_O_2_), oxygen (O_2_), and ozone (O_3_) [1,2,3]. Heterogeneous photocatalysis is one of the most advantageous and widely studied technologies of AOPs [1,4,5,6,7].

The preferential use of nanostructured photocatalysts in AOPs for the degradation of various organic and inorganic water pollutants can be attributed to their environmentally benign properties, including high thermal stability, biological and chemical inertness, and low cost and toxicity [5,8,9,10,11,12,13], mineralizing to water, CO_2_, and minerals. On the other hand, the application of photocatalysts in powder form has been related to technical and practical problems, including the time- and energy-consuming separation/recovery of the photocatalyst from the treated solution [4,14,15,16,17], the mass transfer limitations, and the insufficient irradiation of the photocatalysts in the slurry [18,19,20,21,22] emanating from shielding effects of the suspended solid leading to significantly increasing the operational and capital costs due to the use of high power UV irradiation sources and stirring equipment, as well as the design of more complex processes for the effective separation for the photocatalyst’s retrieval downstream.

A few studies have focused on the effective immobilization of the photocatalysts within various light transparent polymeric matrices with asymmetric pore structure, predominantly structured in beads and fibers [23,24]. In the current work, alginate, a natural, stable gel-forming, linear biopolymer extracted from brown algae, was employed as template to develop novel structured photocatalytic systems in the shape of ceramic beads using commercially available photocatalysts via cross-linking technique. Alginate beads as supporting matrixes for photocatalysts are good candidates for cationic dye adsorption [14,17] and degradation [25], heavy metals and pigments removal [26,27,28], and release of effective antibacterial agents [29].

The most important problem involved in the use of polymeric matrices as shaping agents of the photocatalysts is their potential degradation under the experimental conditions used (continuous UV irradiation) [22,30] and the oxidative action of the photocatalyst, deteriorating the mechanical stability of the material. To overcome these problems, an additional step of pyrolysis/sintering was applied. This treatment removes the structuring polymer, yielding ceramic structures with not only enhanced mechanical stability, but also excellent resistance to attrition. Different protocols of pyrolysis and their effect on the resulting photocatalysts’ performance and stability were investigated.

Furthermore, the novelty of this approach lies in the fact that shaping, structuring, and doping of the photocatalyst with copper nanoparticles (NPs) happens simultaneously. This is achieved by taking advantage of the alginate’s cross-linking ability [15,31,32,33] and metal sorbing capacity [26,34,35] and results in an enhancement of the system’s photocatalytic efficiency. Batch experiments were employed for the assessment of the photocatalytic performance and stability of the developed photocatalytic systems under near-UV irradiation (300–400 nm) using a model pollutant (Methyl Orange, MO), a typical azo dye, whereas their mechanical properties were also examined after several photocatalytic cycles.

## 2. Materials and Methods

All reagents were of analytical grade and used after no further purification. Sodium alginate (SA) was supplied by Sigma-Aldrich Chemie GmbH and TiO_2_ (P25) was supplied by Degussa AG. Glutaraldehyde was purchased from Acros Organics (Geel, Belgium), NaOH and Cu(NO_3_)_2_·3H_2_O from Merck KGaA (Darmstadt, Germany), citric acid from Riedel-de Häen (Seelze, Germany), and ethanol was supplied by VWR International Ltd. (Lutterworth, UK). For the evaluation of the photocatalytic efficiency of the prepared materials, Methyl Orange (MO, 99%, Sigma-Aldrich, St. Louis, MO, USA) dye was used as the water pollutant.

### 2.1. Fabrication of Beads

The dope solution for the preparation of the ceramic TiO_2_ beads (150 mL) was prepared according to the procedure described in our previous work [36]. The TiO_2_/alginate ratio was defined after repeated trials aiming to optimize the material’s mechanical stability. The dope was outgassed for 1 h and then injected dropwise into the coagulation bath using a peristaltic pump and a coaxial bead generator (NiscoVar J1, Nisco Engineering AG, Zürich, Switzerland), depicted in Figure 1. Two different coagulation agents were used to derive the precursor molds of the ceramic beads.

In the first case, 0.8 mL/min of the dope solution was injected into a 500 mL Cu^2+^ solution of 2% (*w*/*v*) (cross-linking solution) under gentle stirring at room temperature. The resulting bead-shaped molds were left for 24 h in the copper solution under stirring to equilibrate. The beads were subsequently filtered, washed several times with ultrapure water, and left to dry at room temperature.

In the second case, 2 mL/min of the dope solution was injected into a 500 mL solution of ethanol, glutaraldehyde, and HCl 5N (90.16 %vol, 8 %vol, and 1.84 %vol), under gentle stirring. The formed beads were filtered and washed several times with ethanol. To further enhance the photocatalytic performance, copper decoration was applied. Specifically, the polymeric molds prepared using glutaraldehyde as the cross-linking agent, after filtration, were dipped into a 300 mL Cu^2+^ ethanolic solution of 1% (*w*/*v*) under gentle stirring at room temperature and left to dry at room temperature.

Finally, the ceramic beads were produced from the dried polymeric molds by calcination/sintering in air or by pyrolysis/sintering under N_2_ flow in the isothermal zone (15 cm) of a tubular furnace (Carbolite CTF furnace (Carbolite Furnaces, Sheffield, UK) equipped with a Eurotherm controller). The heating and cooling rates of 5 °C/min and 2 °C/min, respectively, the duration of the isothermal step (6 h), and the final temperature (600 °C) were common to all the thermal treatment procedures. The developed ceramic beads with their nomenclature are summarized in Table 1.

### 2.2. Characterization Techniques

A Jeol JSM-7401F Field Emission Scanning Electron Microscope (Tokyo, Japan) was used for the characterization of the beads’ surface morphology. The system was equipped with Gentle Beam mode (up to 2.0 kV), which enabled the observation of the samples without sputtering. Gentle Beam technology can reduce charging and improve resolution, signal-to-noise ratio, and beam brightness.

The pore structural assets of the produced ceramic beads were evaluated by nitrogen adsorption-desorption isotherms at 77 K, using an automated volumetric system (AUTOSORB-1 Quantachrome Instruments, Boynton Beach, FL, USA). Prior to the measurements, the samples were outgassed at 180 °C for 48 h, under high vacuum, achieved by means of a turbomolecular pump. The specific surface area (S_BET_) was calculated by the Brunauer–Emmett–Teller (BET) method, while the pore size distribution was determined by the Barrett–Joyner–Halenda (BJH) based on a modified Kelvin equation and the Non-Local Density Functional Theory (NLDFT) methods. The average pore size (nm) was calculated from the porosimetry results as 4000·TPV/S_BET_ [37] and the mean particle size (nm) as 6000/S_BET_·d_sample_ [38], where TPV is the total pore volume and d_sample_ is the density of the sample, assuming cylindrical pore geometry of the empty space between the packed, spherical TiO_2_ nanoparticles.

A Rigaku R-AXIS IV Imaging Plate Detector (Rigaku Corporation, Tokyo, Japan) mounted on a Rigaku RU-H3R Rotating Anode X-ray Generator (operating at 50 kV, 100 mA, nickel-filtered Cu Ka1 radiation, Rigaku Corporation, Tokyo, Japan) was used for the recording of the XRD diffraction patterns. The samples were sealed in Lindemann capillaries.

### 2.3. Photocatalytic Experiments of the Developed Beads

Evaluation of the photocatalytic efficiency of the prepared ceramic beads was performed via batch experiments. The experimental conditions applied and the photocatalytic reactor used are described in more detail in a previous work [36]. In typical photocatalytic experiments, MO adsorption capacity (dark), photocatalytic oxidation efficiency, and UV irradiation reaction kinetics were determined using 30 mL of a 12 ppm MO solution and 75 mg of ceramic beads in a borosilicate glass cell. The cell was positioned into a black pigmented illumination box (50 × 40 × 30 cm) 5 cm from the UV sources (4x Sylvania GTE F15W/T8 lamps, 350–390 nm; photon flux = 15.5 μmoL/m^2^s), achieving an irradiation intensity of 0.5 mW/cm^2^, incident to the top surface of the solution. Before irradiating, the solution underwent magnetic stirring in the dark to establish the time needed to achieve the adsorption equilibrium for all samples. After equilibration, the initial concentration of the solution was measured (C_0_ = C_e_) and was used to deduce the contribution of adsorption in the dark and determine the initiation time (t = 0) for the photocatalytic reactions. A photolysis experiment verified a negligible effect on the MO pollutant concentration upon irradiation with UV light. Homogeneity was attained by vigorous stirring of the solutions throughout the photocatalytic experiments. Furthermore, vigorous stirring was exploited as a means to obtain information on the mechanical stability of the beads, by comparing the weight of the recovered dried samples after each experimental cycle, to that of the starting material.

A Hitachi U-3010 UV-visible spectrophotometer (Tokyo, Japan) was used for determining the MO concentration in the aliquots obtained from the aquatic solutions, taking advantage of the absorption/concentration linear relationship over the respective characteristic absorbance peaks based on Beer–Lambert’s law (464 nm for azo group and 271 nm for aromatic ring).

Moreover, the photocatalytic performance of the ceramic beads was examined with four photocatalysis/regeneration cycles, aiming to verify the performance stability and resistance to attrition of the structured photocatalyst. Upon completion of each photocatalytic test, the sample of beads was easily reclaimed and rinsed twice with ultrapure water. After drying, the sample was subjected to the successive photocatalytic cycle.

MO amount adsorbed onto the beads at equilibrium (q_e_, mg/g) was calculated by:q_e_ = (C_0_ − C_e_) × V/w,
where C_0_ represents the initial concentration (mg/L MO), C_e_ the concentration at equilibrium (mg/L MO), V the solution volume (L), and w the adsorbent weight (g). Finally, the photodegradation efficiency (%) was calculated from:Rejection (%) = (C_e_ − C_t_)/C_e_ × 100%,
where C_t_ represents the concentration at any time during the experiment (mg/L MO).

## 3. Results and Discussion

### 3.1. Materials Characterization

#### 3.1.1. Morphological Analysis of the Ceramic Beads

Figure 2a,e show a macroscopic view of the B1_600 and B2_600 ceramic beads, respectively, outlining their perfectly spherical shape with a diameter of 1–2 mm. The extensive shell roughness of B1_600 uncovers the existence of large flakes with an average size of 2–3 microns. These flakes constitute the salient structural unit of the overall bead and consist of copper oxide- and metallic copper-decorated TiO_2_ NPs aggregates, kept together by the carbonized alginate phase acting as a binder. In contrast to B1_600, the shell of B2_600 appears as an even and smooth surface. Apparently, gelation of the alginate precursor plays an important role in the structural characteristics of the final ceramic material. Gelation with divalent cations, such as copper ions, results in the formation of three-dimensional ionotropic gels via interactions of Cu^2+^ ions with the carboxyl moieties of adjacent G-blocks, a structure known as the egg-box model. Upon contact of the aqueous alginate/TiO_2_ droplet with the Cu^2+^ solution, Cu^2+^ ions start diffusing radially into the droplet. During this diffusion process, fluid instabilities are said to arise from the friction forces involved in the contraction of alginate polymer chains to the newly forming gel front causing anisotropy. Moreover, as gelation proceeds fast, polymer chains are pulled towards the surface of the bead resulting in differences in density between its surface and its core [39,40,41,42,43,44]. This resulting anisotropy will increase the probability of the encapsulated TiO_2_ NPs to come in closer proximity and form larger aggregates. On the other hand, cross-linking with glutaraldehyde, in the presence of HCl, the oxygen lone-pair electrons on the –OH group from alginate can readily attack the polarized carbonyl groups, resulting in rapid nucleophilic addition to the aldehyde, forming monacetal. The adjacent aldehyde in the glutaraldehyde molecule follows a similar reaction mechanism to react with hydroxyls of alginate polymer in near proximity resulting in a bis-acetal cross-linking structure between two uronic acid monomers from adjacent polymer chains leading to intermolecular cross-linking and forming a three-dimensional gel structure [45]. This reaction together with the presence of H^+^ ions from HCl promoting isotropic alginate gelation, leads to the formation of a homogeneously distributed polymer network supporting the entrapped TiO_2_ particles. Hence, cross-linking with glutaraldehyde prior to exposure to Cu^2+^ will result in a more homogeneous precursor, as seen in the SEM micrographs.

However, even though there are differences in the morphological characteristics of the precursor depending on the way gelation proceeded, both processes will result in bringing the entrapped TiO_2_ NPs in close vicinity with copper species as the metal binding sites are distributed evenly all across the polymeric matrix. Hence, after the carbonization procedure, Cu and/or CuO nanoparticles will form in the final ceramic structure, in good dispersion and in firm contact with the TiO_2_ NPs, yielding structured photocatalysts augmented with well-distributed NPs.

Comparison between B1_600 (Figure 2b) and B2_600 (Figure 2f) samples shows that the direct use of Cu^2+^ as the cross-linking agent (B1_600) results in the formation of densely structured, large, and elongated aggregates of TiO_2_ NPs of a few microns in size (2–3 μm) with a quite large average inter-aggregate space in the range of 1–2 μm. On the other hand, aggregates in B2_600 do not exceed the size of 1 micron, with smaller inter-aggregate space up to 500 nm. Furthermore, comparing Figure 2d,h, one can observe that depending on the approach used for bead preparation, TiO_2_ nanoparticles exhibit different distribution densities into the ceramic matrix of the beads with dimensions ranging from 30 to 74 and 22 to 40 nm for B1_600 and B2_600 samples, respectively.

Finally, Energy Dispersive Spectroscopy (EDS) analysis was employed to define the samples’ composition, as well as their dispersion characteristics (data not shown). In B1_600 and B2_600 samples, average copper content reached ~3.0% (mean value from 10 different areas). This value was expected based on the alginates’ sorption capacity documented in the literature [26], and previously reported in prepared ceramics in the form of fibers [36].

#### 3.1.2. LN_2_ Porosimetry of Ceramic Beads

The textural and structural pore properties of the developed materials were explored by nitrogen porosimetry at 77 K (Figure 3a). In addition, porosity, total pore volumes (TPV) at 0.99 P/P_0_, and BET surface area for Degussa P25 TiO_2_ (starting material, for comparison), as well as for the obtained bead samples are shown in Table 2. At 0.97 P/P_0_, pores formed due to the close-packing of spherical TiO_2_ NPs, with a high number of contact points, accounting for the cumulative volume of adsorbed liquid nitrogen. On the other hand, in the range of 0.97 to 0.99, the adsorbed volume can be attributed to the empty space between the larger TiO_2_ NPs aggregates, which corresponds to macropores. The pore size distribution (PSD) of the ceramic beads was determined from the LN_2_ isotherms desorption branch via the BJH and NLDFT methods. For the NLDFT method, an equilibrium kernel for silica as adsorbent and N_2_ 77 K as adsorbate was used.

A first conclusion drawn from the results of LN_2_ porosimetry at 77 K is the detrimental effect of calcination; sample B1_600_air exhibits the lowest TPV and surface area amongst all other ceramic samples including pristine Degussa P25. The presence of air not only burns off residual carbon, which is partly responsible for the final porous structure, but also results in sintering of the TiO_2_ NPs leading to an almost complete collapse of the porous network up to 25 nm (Figure 3b), hence rendering it unusable for the intended application.

On the other hand, for the N_2_ treated samples, a complex pore structure is revealed, with the existence of a broad range of pore sizes, with B1_600 beads exhibiting narrower pore size distribution than B2_600, as depicted in Figure 3b. This difference reflects the different process steps involving the preparation of the precursor molds. As mentioned previously, the polymeric precursors of the ceramic B2 beads were prepared using glutaraldehyde as the cross-linking agent and were subsequently contacted with a Cu(NO_3_)_2_ solution for copper decoration via Cu^2+^ uptake. On the other hand, for B1 beads, Cu^2+^ served a dual purpose: a gelling agent/cross-linker and copper decoration, resulting in quite different pore structure properties. Indeed, while B2_600 exhibits a type II N_2_ adsorption/desorption isotherm, B1_600 isotherm is a typical type IV isotherm, with an H1-type hysteresis loop, which according to IUPAC classification, characterizes mesoporous materials.

Unlike the air-treated sample, the samples treated under inert gas atmosphere avoid sintering phenomena and the TiO_2_ NPs mostly retain the structure of the parent material (Degussa P25), while their surface gets coated with residual carbon produced from the pyrolysis of the polymeric mold. Residual carbon, apart from acting as a binder, also contributes to the development of specific structural features by partially filling the interparticle space, yielding materials with different morphological and pore structural characteristics depending on the cross-linking procedure. B1_600, produced using a less homogeneous mold due to the different gelling procedure, exhibited lower TPV and specific surface area compared to B2_600 beads (Table 2). The NLDFT analysis (Figure 3b-top) shows that while in the B2_600 beads, smaller pores that also existed in the parent Degussa P25 were retained and there was also a slight increase of larger inter-aggregate mesopores, in the B1_600 beads, smaller pores were greatly reduced, while larger ones were formed in the inter-aggregate space, because of the effects from the presence of carbon and metal nanoparticles formed during pyrolysis. In such composite materials with complex pore structures, bottleneck pores and constrictions of larger pores by smaller ones may occur, being ultimately responsible for the final porous structure. Mean particle sizes were also in agreement with the results from the morphological analysis.

In any case, even though morphological analysis (Section 3.1.1) revealed inter-aggregate spaces of 1–2 μm, LN_2_ porosimetry is only applicable in pore size ranges of 0.35 to 500 nm, so the analysis in this section refers solely to the intra-aggregate space within those larger and densely structured aggregates, as well as the intraparticle pores of TiO_2_ NPs.

### 3.2. XRD Analysis

The XRD diffractograms of the ceramic beads and the Degussa P25 TiO_2_ (for comparison) are depicted in Figure 4. All composite ceramic samples consist of both anatase (the diffraction peaks appeared at 2θ = 25.4°, 37.1°, 38.0°, 38.8°, 48.2°, 54.0° and 55.1°) and rutile phases (the diffraction peaks appeared at 2θ = 27.6°, 36.2°, 41.4°, 44.2°, and 56.7°).

The weight fraction of rutile can be calculated by the diffraction peak intensities of the anatase (101) and rutile (110) phases, using the following equation [46]:w_r_ = 1/(1 + 0.79·I_A_/I_R_),
where w_r_, is the weight fraction of the rutile phase, I_A_, is the anatase (101) plane diffraction peak intensity, and I_R_, is the rutile (110) plane diffraction peak intensity.

Moreover, the mean size of crystallites was derived from Scherrer’s formula:

t = K·λ/Β·cosθ with K = 0.89 (shape factor) and λ = 0.154056 nm (X-ray wavelength).

The anatase to rutile phase ratios and the mean size of crystallites for all samples are shown in Table 3. Even though pyrolyzed B1_600 and B2_600 samples retained the anatase to rutile phase ratio of the parent material, in the calcined B1_600_air sample, the anatase phase content was significantly decreased (~40 wt.%) and was partially transformed to rutile. Moreover, the presence of carbon was confirmed for the pyrolyzed B1_600 and B2_600 samples treated under inert atmosphere, in contrast to the calcined B1_600_air, where no carbon phase exists. The presence of residual carbon, which is also justified by Raman analysis (Supporting Information), in the pyrolyzed ceramic bead samples inhibited the thermally induced aggregation and consequently prevents the anatase-rutile phase transformation.

Furthermore, even though the copper content of the ceramic beads was ~3% for all samples, peaks corresponding to copper species can be seen only in the XRD diffractograms of B1_600 and B1_600_air samples.

In the B1_600 sample, diffraction peaks corresponding to zero-valent metallic Cu indicate that, during pyrolysis, copper ions bound to the carboxyl groups and in the close vicinity of the hydroxyl groups of the uronic acid residues, underwent reduction as resembling the first step of the polyol process [39,47]. From the width of the peak at 43.3°, the mean size of the metallic copper NPs of B1_600 was calculated at 24 nm using Scherrer’s formula. Metallic copper is also expected to be present in the other pyrolysis derived sample (B2_600) and the absence of the corresponding diffraction peaks indicates very high dispersion of the metallic nanoparticles in the material. The difference in dispersion of copper species between B1_600 and B2_600 beads can be attributed to the different cross-linking approaches employed for the preparation of their precursors. The different structural characteristics of the precursor gels in terms of anisotropy and homogeneity promote the formation of larger metallic particles in the case of B1_600 beads not so highly dispersed into the final ceramic sample. On the other hand, for the B1_600_air sample calcined under air, Bragg reflections corresponding to CuO (111) can be identified in the XRD diffractogram, indicating the formation of copper oxide in the presence of air during calcination. Employing Scherrer’s formula at 38.8°, the CuO NPs mean size was calculated at around 21 nm. The presence of CuO could also be the reason for the promotion of the anatase-rutile phase transformation in B1_600_air, as it has been documented that the CuO additive could noticeably enhance the transformation by reducing the onset temperature of the phase transition [48].

### 3.3. MO Adsorption and Photocatalytic Performance of Beads

Adsorption and photocatalysis quite commonly exhibit synergistic effects especially when high efficiency adsorbents (e.g., activated carbon) reside in the structured photocatalyst’s pores [36]. In that regard, both photocatalysis and adsorption were investigated in order to obtain an estimation of the beads’ overall performance.

In this work, the adsorption and photocatalytic performance of three bead samples was evaluated in a 12 ppm MO solution (nomenclature for the samples in Section 2.2). As observed in Figure 5a, B2_600, with its enhanced BET and TPV compared to the other samples, showed the highest (3.0 mg/g) adsorption capacity determined from the attenuation of the main MO absorbance peak (464 nm). The lowest adsorption capacity (0.1 mg/g) was observed for the calcination derived sample (B1_600_air). The enhanced MO adsorption capacity achieved by the pyrolysis-derived samples is a consequence of the residual carbon and the dispersive interaction mechanisms driven by the π-π dispersion interaction between the adsorbate and the residual carbonaceous phase or the electron donor-acceptor complex mechanism [49,50]. Photocatalytic performance can be enhanced by the presence of the carbonaceous phase adsorbent in two ways. Firstly, adsorption results in increased concentration of pollutants in the vicinity of the photocatalyst, as carbon is usually interfacing or very close to the TiO_2_ NPs’ surface and secondly, by interacting with the adsorbed molecules, the carbonaceous phase can contribute to bond relaxation [51] making them susceptible to photocatalytic cleavage.

Obviously, higher BET surface area and pore volume significantly contribute to enhanced MO adsorption capacity of these carbon containing samples, since the MO adsorption capacity converges with their TPV and BET ranking. However, differences related to the residual carbon’s oxygenated groups speciation and percentage [36] can result in different MO adsorptivity between B1_600 and B2_600 beads. Hence, pore properties and NPs content may not be the sole factors contributing to the higher MO adsorptivity of B2_600 compared to B1_600 samples [36].

The photocatalytic efficiency of the samples under study is presented in Figure 5b. The plot shows a very good accordance between their ranking with regard to their MO adsorption capacity and MO degradation effectiveness.

Pyrolytic treatment together with the incorporation of a co-catalyst (copper species) alongside TiO_2_ enhances the photocatalytic dye degradation. The enhanced pore volume and surface texture of the samples provide numerous active MO adsorption sites. Usually, as mentioned earlier, the increased photocatalytic efficiency is credited to the pollutants’ local concentration increase due to adsorption on the photocatalyst’s surface, as well as to electron transfer from the semiconductor to the metal nanoparticles, significantly inhibiting electron–hole recombination. In our specific case, the Cu NPs could serve as sink for TiO_2_ generated electrons when under irradiation [36].

Moreover, the reductive path of the photocatalysis via the metallic copper NPs should be considered. The copper NPs incorporated into TiO_2_ receive the photogenerated electrons of the semiconductor, which migrate to its surface and induce the adsorbed MO molecules azo group reductive cleavage. In parallel, the photocatalytic oxidative pathway is accomplished, as the photogenerated holes oxidize surface-adsorbed water and produce highly oxidizing hydroxyl radicals (OH^•^) together with oxygen. The latter could react with the chromophore group (–N=N–) conveying electrons from the photocatalyst’s surface and generating hydroxylated products.

As clearly observed in Figure 5a, the adsorption capacity of Degussa P25 TiO_2_ is inferior to the pyrolytic treated beads proving that there is no chemical interaction between MO molecule and reference photocatalyst. On the other hand, the enhanced adsorption capacity due to extensive pore network yields more efficient beads photocatalysts affiliating severely with the probe molecule owing to the residual carbon existence. Regarding the photocatalytic performance, it is apparent that Degussa P25 TiO_2_ presents clearly better photocatalytic efficiency than the photocatalytic beads, especially after 60 minutes of UV irradiation, using the same loading and irradiation conditions (Figure 5b). This deteriorated performance was also observed in previous studies [52] due to inappropriate irradiation. As the commercial photocatalyst is shaped into beads, the TiO_2_ nanoparticles present at the inner part of the bead do not contribute significantly to the overall performance, due to limited irradiation. In contrast, the attribution of such inner nanoparticles to adsorption capacity is significant. Despite the decreased photocatalytic performance, the overall MO rejection by the beads in the combined adsorption-photocatalytic process exceeds the 80%.

To prove the efficiency of the proposed engineered photocatalysts, the obtained results are presented in comparison with the photocatalytic performance reported in the literature for Degussa P25 TiO_2_-based photocatalytic systems, in the form of film/coating or slurry (Table 4). At this point, it should be mentioned that the light intensity is a decisive parameter when attempting to benchmark photocatalytic systems, as it rules the amount of the generated electron-hole pairs [53,54]. Thus, apart from the amount of the catalysts and the concentration of the solutions used, Table 4 also contains details about the irradiation parameters applied in every study. It is remarkable that the photocatalytic efficiency of the beads, tested at low light intensity conditions, is equal or greater than the ones documented in the literature for films/coatings, tested under similar experimental conditions. On the other hand, despite the better performance of slurry systems, there is always a trade-off between their high photocatalytic efficiency and the difficulties in recovering them effectively from the solution and irradiating them efficiently in the slurry. Although the increased photocatalytic performance of the shaped photocatalysts can also be attributed to the beneficial role of the copper NPs, the proposed methodology can provide structured photocatalysts with inherited enhanced activity. Further studies will be focused on the optimization of photonic efficiency of the beads and the evaluation of their performance in a fixed or a fluidized-bed reactor.

### 3.4. Stability of the Photocatalytic Performance of Beads

The shape and size of the photocatalytic composite beads permits their facile separation from the treated dye solution and their reuse in a fresh solution of MO pollutant, thus providing the opportunity to explore their stability. To this end, after regeneration of the beads (procedure described in Section 2.3), the MO adsorption and photocatalytic degradation experiments were repeated for four successive cycles. When the cycle was completed, beads were easily separated from the MO solution and washed twice with ultrapure water, before being reused in the subsequent cycle in fresh (C_0_ = 12 ppm) MO solution.

During the last cycle of experiments, B2_600 beads still exhibited higher adsorption capacity and photocatalytic efficiency than B1_600 beads. The amount of MO adsorbed and the MO rejection efficiency are summarized in Table 5, in comparison to the performance obtained during the first cycle (values in parentheses). Even though B1_600 beads proved more effective in preserving their adsorption capacity after four experimental cycles (3% drop in adsorption capacity against 25% for B2_600), B2_600 beads are endowed with higher stability of their photocatalytic performance, maintaining 88.5% of its initial value against 85% for B1_600. This asset is attributed to the higher Cu NP dispersion in the B2_600 sample. Overall, the beads exhibit only a slight degradation of their photocatalytic efficiency, keeping 87% (average) of their initial rejection values.

For B1_600 and B2_600 samples, the mass loss percent was 0.21% and 1.19%, respectively, indicating that B1_600 beads exhibit slightly better resistance to attrition compared to B2_600. In any case, the weight loss can be considered minimal, indicating the good mechanical properties of the prepared materials. However, the increased mass loss observed for the B2_600 could probably be attributed to loss of residual carbon, which would explain the larger drop in adsorption capacity after the fourth cycle compared to B1_600.

## 4. Conclusions

In this work, two cross-linking approaches were employed for the preparation of Cu-alginate/TiO_2_ molds followed by two thermal treatment protocols (pyrolysis and calcination) for the preparation of Cu/TiO_2_ ceramic photocatalytic beads. Residual carbon present in the pyrolysis-derived samples was found not only to enhance the mechanical properties serving as binding material between the sintered TiO_2_ NPs, but also to increase the adsorption capacity due to the carbon’s extended pore structure.

The use of glutaraldehyde as the cross-linker, followed by Cu^2+^ adsorption from the precursor molds and pyrolytic treatment at 600 °C for 6 h yielded the most effective beads with 3.0 mg/g MO adsorption capacity and a photocatalytic efficiency of about 50%, reaching an overall MO rejection from water of approximately 80% at short contact times of less than 3 hrs. Their photocatalytic performance is contributed to by their extended pore structure and their decoration with zero-valent copper nanoparticles. The latter trigger an additional MO degradation mechanism along with the conventional one of oxidation through the hydroxyl radicals produced by water splitting on the photogenerated semiconductor’s holes. Hence, copper nanoparticles act as a sink for the photogenerated electrons. Moreover, the bead-shaped photocatalysts show good mechanical stability and resistance to attrition, retaining in general their photocatalytic efficiency after at least four successive photocatalytic cycles.

In summary, the bead-shaped photocatalysts possess enhanced mechanical properties and their photocatalytic activity increases significantly in the presence of copper NPs. Ongoing dynamic experiments under continuous flow aim to corroborate their applicability in fixed or fluidized bed photocatalytic reactors (FBPR) and their feasibility for industrial scale wastewater treatment processes.

## Figures and Tables

**Figure 1 materials-15-00326-f001:**
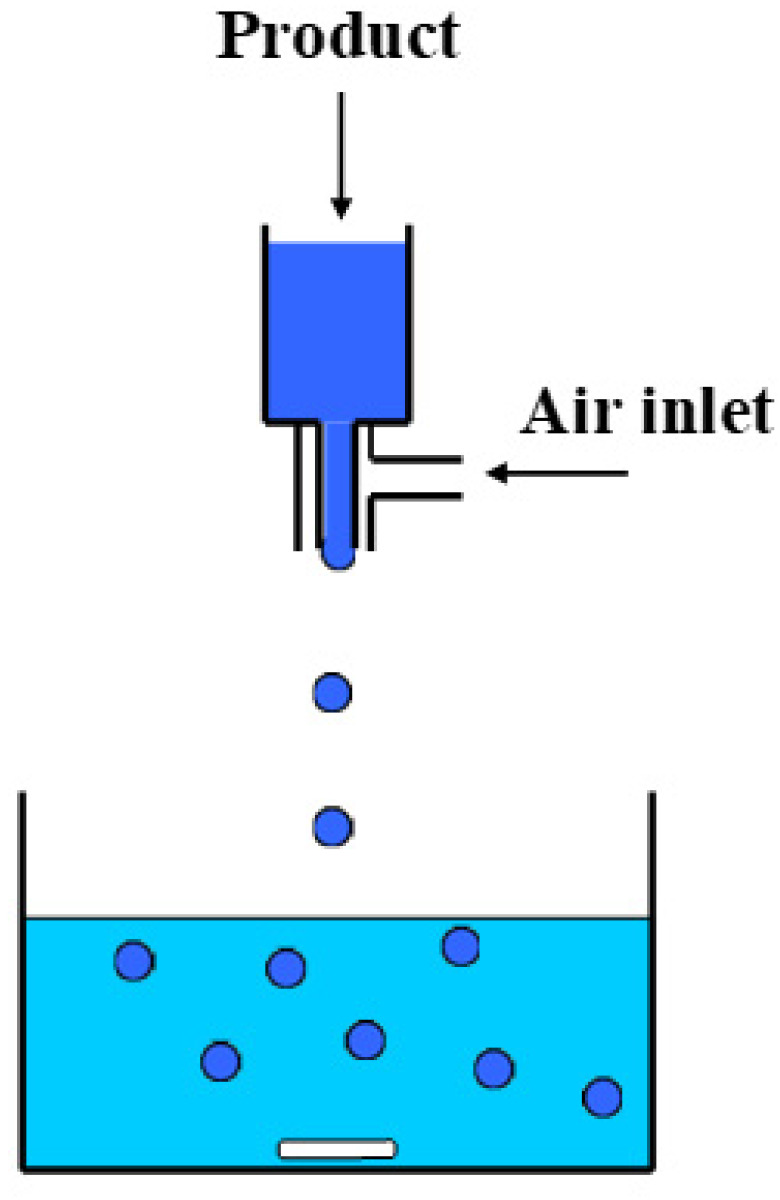
Principle of coaxial airflow bead generator.

**Figure 2 materials-15-00326-f002:**
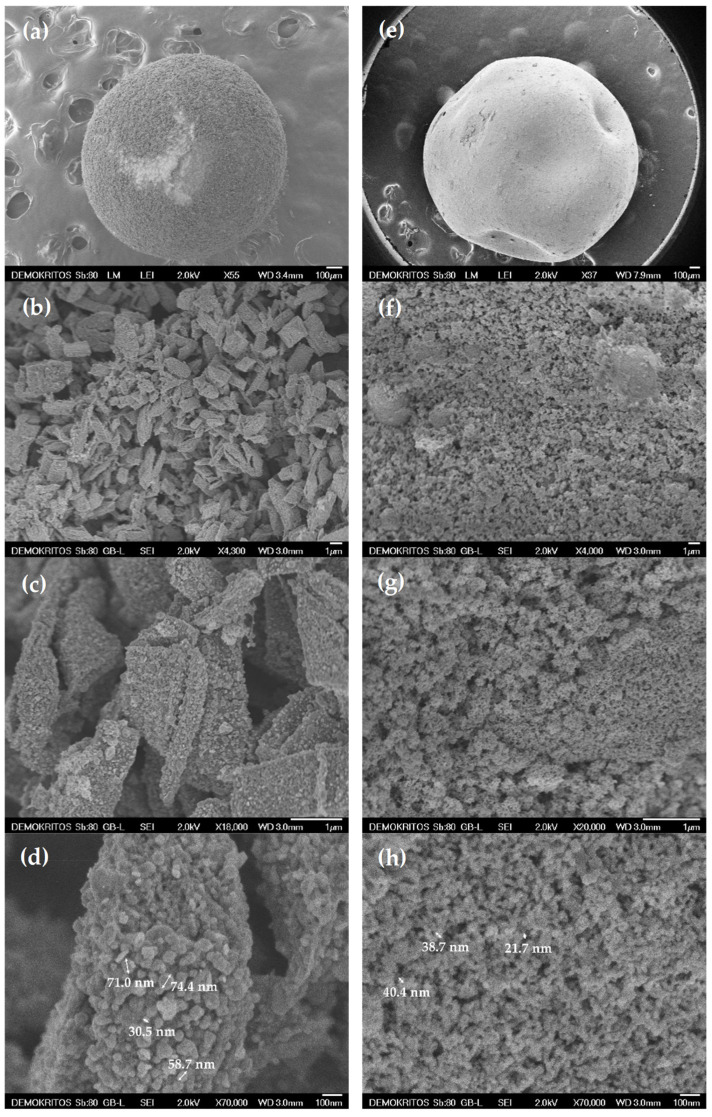
SEM images of the external surface of (**a**–**d**) B1_600 and (**e**–**h**) B2_600 samples.

**Figure 3 materials-15-00326-f003:**
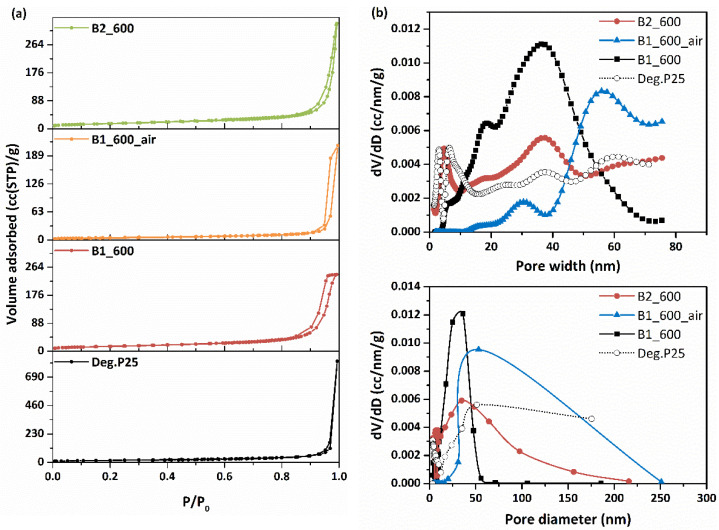
(**a**) N_2_ adsorption (77 K) of the prepared ceramic beads in comparison to N_2_ adsorption of Degussa P25 TiO_2_. (**b**) Pore size distributions calculated from the desorption branch of ceramic beads employing the BJH (bottom) and NLDFT (top) methods.

**Figure 4 materials-15-00326-f004:**
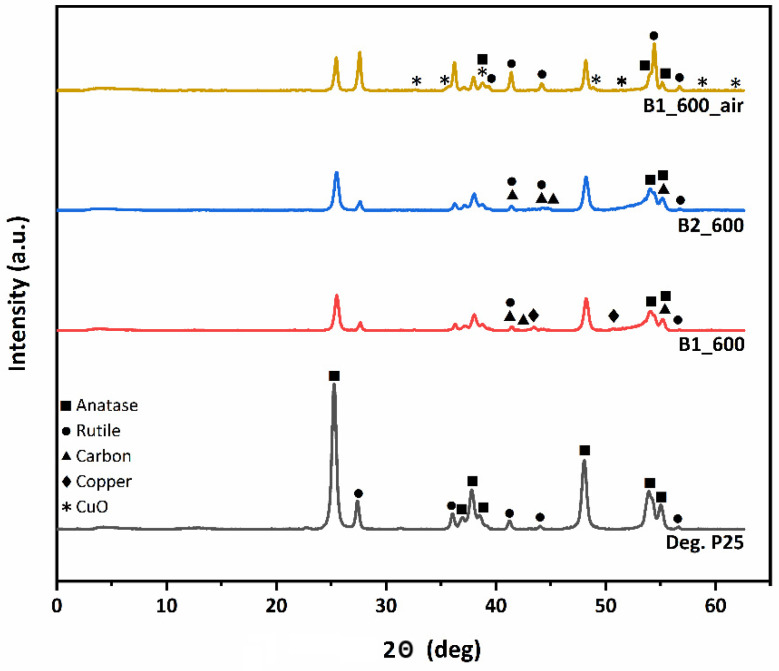
XRD patterns of the ceramic beads compared to Degussa P25 TiO_2_.

**Figure 5 materials-15-00326-f005:**
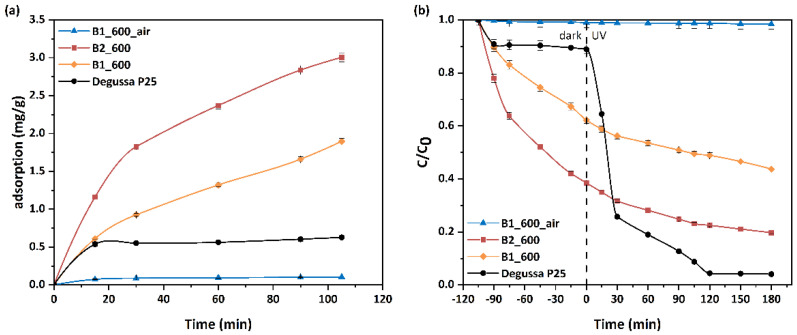
(**a**) Adsorption kinetics curves for MO adsorption at a concentration of 12 ppm of fresh beads samples in comparison to MO adsorption of Degussa P25 TiO_2_ (pH = 6, 25 °C). (**b**) Kinetics of MO photocatalytic degradation for all samples in comparison to MO degradation of Degussa P25 TiO_2_ (UV irradiation, C_0_ = 12 ppm, pH = 6, 25 °C).

**Table 1 materials-15-00326-t001:** Various protocols of calcination and pyrolytic/sintering treatment of beads.

Code Sample	Cross-Linker		Process	Τ	Isothermal Step
	Glutaraldehyde	Cu^2+^		(°C)	(h)
B1_600	−	+	pyrolysis	600	6
B1_600_air	−	+	calcination	600	6
B2_600	+	+	pyrolysis	600	6

**Table 2 materials-15-00326-t002:** Pore characteristics and particle sizes of all samples.

Sample	TPV ^1^	S_BET_	Porosity ε	d_mean_ ^2^	d_BJH_ ^3^	D_particle_ ^4^
	(mL/g)	(m^2^/g)	(%)	(nm)	(nm)	(nm)
B1_600	0.374	53.8	59.3	27.8	36.1	29.5
B1_600_air	0.228	19.4	47.1	46.9	53.1	79.1
B2_600	0.513	57.2	66.7	35.9	34.8	26.9
Deg. P25	0.269	62.0	51.2	17.3	51.2	24.8

^1^ Total pore volume at 0.99. ^2^ Mean pore size d_mean_ as 4000·TPV/S_BET._
^3^ Pore size determined from the pore size distribution (PSD) using the Barrett-Joyner-Halenda (BJH) method based on a modified Kelvin equation of the N_2_ desorption branch. ^4^ Mean particle size D_particle_ as 6000/S_BET_·d_sample_ (d_sample_ density in g/cm^3^).

**Table 3 materials-15-00326-t003:** Weight fraction of anatase and rutile phases and mean size of crystallites for all samples.

Sample	Anatase (nm)	Rutile (nm)	w_a_ (%)	w_r_ (%)
B1_600	17.8	23.9	78.6	21.4
B1_600_air	24.1	25.2	39.9	60.1
B2_600	17.4	23.8	80.0	20.0
Deg. P25	17.3	23.6	81.0	19.0

**Table 4 materials-15-00326-t004:** Photocatalytic performance of Degussa P25 TiO_2_-based photocatalysts.

Form	Catalyst Amount (g/L)	Light Intensity (mW/cm^2^)	Results	Reference
powder	0.8	9.4	R ≈ 65% (15 ppm, 3 h)	[55]
powder	1	11.1	R ≈ 60% (15.6 ppm, 3 h)	[56]
powder	0.16	7.75	R ≈ 50% (10 ppm, 1 h)	[57]
powder	0.3	n/a	R ≈ 58% (20 ppm, 1.5 h)	[58]
coating *	1	25.3	R ≈ 55% (12 ppm, 3 h)	[59]
coating	0.2	1.73	R ≈ 29% (9.8 ppm, 3 h)	[60]
powder	2.5	0.5	R = 95.4% (12 ppm, 3 h)	This work
beads	2.5	0.5	R = 48.8% (12 ppm, 3 h), R = 80.3% for the overall process at the same conditions	This work

* Titania PC500 (anatase: >99%, specific surface area 350 to 400 m^2^/g, crystallites mean size = 5–10 nm).

**Table 5 materials-15-00326-t005:** Amount of MO adsorbed (mg/g) and final MO rejection (%), due to photocatalytic degradation, after 4 successive cycles (C_0_ = 12 ppm, pH = 6, 25 °C) compared to the data obtained from the first cycle (values in parentheses).

Sample	q_e_	R
	(mg/g)	(%)
B2_600	2.24 (3.00)	43.2 (48.8)
B1_600	1.85 (1.90)	25.3 (29.7)

## Data Availability

Not applicable.

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
