# Peer review of "Engineering Commercial TiO2 Powder into Tailored Beads for Efficient Water Purification"

_materials, 2022, doi:10.3390/ma15010326_

Round 1

Reviewer 1 Report

The authors report on the photocatalytic decomposition of methyl-orange on TiO2/alginate-based composite bead-shaped catalysts. The catalytic process was investigated in the usual way to determine the decomposed amount of the MO by UV-VIS spectrometry taking into account the adsorption correction. The application of the P25 Degussa TiO2 in every way is getting far from the novelty. Moreover, the decomposition of the methyl-orange dye is an even more outdated selection to demonstrate any kind of photocatalytic activity. 

Some questions and remarks to the authors for further clarification:

-Please improve the scaling bar visibility in Figure 2

-Unify the outlook of the Table series (Table 1 has a different format)

-Could you confirm the carbon accumulation in the surface layers by XPS which was mentioned on page 6?

In the manuscript, in the characterization chapter, the authors used the P25 as a reference, that comparison is the most relevant in the real photocatalytic experiments. Therefore in Figure 5. the P25 must be indicated as a reference.

After the adsorption step, the authors should clarify the C0 in every case.  How does proportionately the 12 ppm concentration after the no illumination step?

The paper is well-structured, the selected results are appropriate, the quality of the data is barely acceptable, the presented data on the decomposition is less convincing. However, I recommend that the authors should consider my remarks summarized below and use them to further improve their manuscript.

Author Response

Reviewer #1: The authors report on the photocatalytic decomposition of methyl-orange on TiO2/alginate-based composite bead-shaped catalysts. The catalytic process was investigated in the usual way to determine the decomposed amount of the MO by UV-VIS spectrometry taking into account the adsorption correction. The application of the P25 Degussa TiO2 in every way is getting far from the novelty. Moreover, the decomposition of the methyl-orange dye is an even more outdated selection to demonstrate any kind of photocatalytic activity.

Some questions and remarks to the authors for further clarification:

- Please improve the scaling bar visibility in Figure 2.

Based on reviewer’s suggestion the Figure 2 was amended appropriately.

- Unify the outlook of the Table series (Table 1 has a different format).

As suggested by the reviewer, the Table 1 was revised appropriately.

- Could you confirm the carbon accumulation in the surface layers by XPS which was mentioned on page 6?

The first indication of the presence of residual carbon was due to the dark-grey colour of the samples thermally treated under inert atmosphere. The presence of carbon was also verified by Raman analysis. Based on the discussion of Raman analysis presented in section S1 of the Supporting Information (pages 1-2), it is clear that all samples treated under inert atmosphere contain residual carbon with similar structure and only slight differences in the ID/IG values for B1_600 and B2_600 samples (0.43 and 0.40, respectively) were observed.

- In the manuscript, in the characterization chapter, the authors used the P25 as a reference, that comparison is the most relevant in the real photocatalytic experiments. Therefore in Figure 5. the P25 must be indicated as a reference.

Based on reviewer’s suggestion, reference Degussa P25 was added in Figure 5 and compared with our developed bead samples (page 12 in the revised manuscript).

- After the adsorption step, the authors should clarify the C0 in every case.  How does proportionately the 12 ppm concentration after the no illumination step?

The reviewer is absolutely correct. The initial concentration for experiments was 12 ppm. However due to the differences in adsorption capacity of the various samples, the initial concentration of the photocatalytic experiment is different for every sample. To avoid any misunderstanding, the Figure 5 and its legend of the manuscript was amended, according to the reviewer’s comment. The new Figur3 5 shows the initial concentrations of the experiments in both the dark (adsorption) and under irradiation (photocatalytic degradation). 

The paper is well-structured, the selected results are appropriate, the quality of the data is barely acceptable, the presented data on the decomposition is less convincing. However, I recommend that the authors should consider my remarks summarized below and use them to further improve their manuscript.

Reviewer 2 Report

I review the paper entitled "Engineering Efficient Photocatalysts into Tailored Bead Shaped Systems for Applied Water Treatment" by George V. Theodorakopoulos et al. In this paper, commercially TiO2 nanoparticles were decorated with Cu by pyrolytic or calcination-sintering procedure using alginate as template, leading to ceramic beads, as an innovative technique. The photocatalytic performance of ceramic beads was conducted in UV, using methyl orange as a model pollutant. Morphological and structural characteristics were studied using LN2 porosimetry, SEM and XRD, while the mechanical properties were assessed during four photocatalysis/regeneration cycles as performance stability and resistance to attrition.  The authors performed a good work, the paper is appealing and deserves to be published in Materials Journal.

As suggestions, I recommend the authors to replace Cu+2 (lines 93,103, 208, 218, 262, 465, inside of Table 1) with Cu2+.

Also, please reorganize the Section 2.2 accordingly to the Results and Discussion Section.

The authors compared the SEM images although they are obtained at different magnification. I recommend the authors, as possible, to have the same magnifications for ceramic beads comparison. In addition, the scale bars from Figure 2 and the size dimension of TiO2 nanoparticles (Fig. 2d and h) should be visible under normal view. According to Section 3.3 MO adsorption and photocatalytic performance of beads, the tests were performed at 464 nm. Please authors to correct the range of UV (line 70). Besides these observations, overall, the manuscript contains the correctly methods and adequate results, which conduct to the satisfactory conclusions.

Author Response

Reviewer #2: I review the paper entitled "Engineering Efficient Photocatalysts into Tailored Bead Shaped Systems for Applied Water Treatment" by George V. Theodorakopoulos et al. In this paper, commercially TiO2 nanoparticles were decorated with Cu by pyrolytic or calcination-sintering procedure using alginate as template, leading to ceramic beads, as an innovative technique. The photocatalytic performance of ceramic beads was conducted in UV, using methyl orange as a model pollutant. Morphological and structural characteristics were studied using LN2 porosimetry, SEM and XRD, while the mechanical properties were assessed during four photocatalysis/regeneration cycles as performance stability and resistance to attrition.  The authors performed a good work, the paper is appealing and deserves to be published in Materials Journal.

As suggestions, I recommend the authors to replace Cu+2 (lines 93,103, 208, 218, 262, 465, inside of Table 1) with Cu2+.

The Cu+2 was replaced with Cu2+ in lines 97, 106, 190, 192, 209, 219, 264, 484 and in Table 1 of the revised manuscript, according to the reviewer’s comment.

Also, please reorganize the Section 2.2 accordingly to the Results and Discussion Section.

The section 2 was reorganized based on the reviewer’s comment.

The authors compared the SEM images although they are obtained at different magnification. I recommend the authors, as possible, to have the same magnifications for ceramic beads comparison. In addition, the scale bars from Figure 2 and the size dimension of TiO2 nanoparticles (Fig. 2d and h) should be visible under normal view.

The Figure 2 was thoroughly amended based on the reviewer’s comment.

According to Section 3.3 MO adsorption and photocatalytic performance of beads, the tests were performed at 464 nm. Please authors to correct the range of UV (line 70).

The range of UV light (line 70 in original manuscript) concerns to the lamps’ emission and not to the range (240-800 nm) of the Hitachi U-3010 UV-visible spectrophotometer employed for the determination of the MO concentrations. Sorry for the misunderstanding. We hope that this point is now clarified.

Besides these observations, overall, the manuscript contains the correctly methods and adequate results, which conduct to the satisfactory conclusions.
